# Could Histamine H1 Receptor Antagonists Be Used for Treating COVID-19?

**DOI:** 10.3390/ijms22115672

**Published:** 2021-05-26

**Authors:** Changbo Qu, Gwenny M. Fuhler, Yihang Pan

**Affiliations:** 1Tomas Lindahl Nobel Laureate Laboratory, Precision Medicine Research Center, The Seventh Affiliated Hospital of Sun Yat-sen University, Shenzhen 518107, China; Changbo.Qu@radboudumc.nl; 2Department of Biochemistry, Radboud Institute for Molecular Life Sciences, Radboud University Medical Center, 6500 HB Nijmegen, The Netherlands; 3Department of Gastroenterology and Hepatology, Erasmus MC-University Medical Center, 3015 CN Rotterdam, The Netherlands; g.fuhler@erasmusmc.nl

**Keywords:** COVID-19, NF-κB signaling, H1 receptor antagonists, treatment, drugs

## Abstract

COVID-19 has rapidly become a pandemic worldwide, causing extensive and long-term health issues. There is an urgent need to identify therapies that limit SARS-CoV-2 infection and improve the outcome of COVID-19 patients. Unbalanced lung inflammation is a common feature in severe COVID-19 patients; therefore, reducing lung inflammation can undoubtedly benefit the clinical manifestations. Histamine H1 receptor (H1 receptor) antagonists are widely prescribed medications to treat allergic diseases, while recently it has emerged that they show significant promise as anti-SARS-CoV-2 agents. Here, we briefly summarize the novel use of H1 receptor antagonists in combating SARS-CoV-2 infection. We also describe the potential antiviral mechanisms of H1 receptor antagonists on SARS-CoV-2. Finally, the opportunities and challenges of the use of H1 receptor antagonists in managing COVID-19 are discussed.

## 1. Introduction

Coronavirus disease 2019 (COVID-19), an emerging respiratory disease caused by severe acute respiratory syndrome coronavirus 2 (SARS-CoV-2), is swiftly leading to global health issues and becoming a pandemic worldwide. It forces much of the world to adopt a lockdown mode, causing enormous economic fallout and human suffering. Most patients with COVID-19 are either asymptomatic or show mild symptoms; however in some cases, patients progress to severe lung injuries and eventually develop multiple organ failure [1,2].

SARS-CoV-2 is a single-stranded, positive-sense RNA virus (++ssRNA) [3]. The SARS-CoV-2 genome possesses an 82% sequence identity to that of SARS-CoV and MERS-CoV. Four structural proteins including spike (S), envelope (E), membrane (M), and nucleocapsid (N) proteins have been identified in SARS-CoV-2. These protein sequences are also highly similar to that of SARS-CoV and MERS-CoV [4]. The viral structural proteins play vital roles in determining the viral life cycle, and thus provide potential therapeutic targets [5]. SARS-CoV-2 engages SARS-CoV angiotensin converting enzyme 2 (ACE2) receptor for entry and transmembrane serine protease (TMPRSS2) for S protein priming. After entering the cell, SARS-CoV-2 is subsequently taken up into endosomes and then fused with lysosomal membranes. Eventually, SARS-CoV-2 virions are released from the cell through exocytosis (Figure 1) [6]. SARS-CoV-2 infection can cause severe respiratory pathologies and lung injuries [7]. The severity of the lung injuries is correlated with the production of a cytokine storm by the macrophages during SARS-CoV-2 infection. High levels of cytokines including IL-2, IL-10, GCSF, IP-10, MCP-1, IL-7, TNF-α, and MIP-1A were observed in COVID-19 patients at high risk of mortality [1]. In parallel, an enhanced concentration of perivascular and septal mast cells was found in post-mortem lung biopsies of COVID-19 [8]. Mast cells synthesize and secrete inflammatory mediators including histamine. The roles of mast cells in SARS-CoV-2 infection have been frequently discussed [9,10,11,12]. Whether histamine released by mast cell activation during SARS-CoV-2 infection contributes to the severity of lung injury remains to be elucidated [13,14].

In most cases, the excess lung inflammation response caused by SARS-CoV-2 is self-competent; however, in some patients, it is unbalanced and non-competent, with age and comorbidities such as arterial hypertension or diabetes being acknowledged as risk factors. As a consequence, these patients require hospitalization and need to be managed appropriately. Considering the alleviation of the inflammatory response and concomitant lung injuries, anti-inflammatory drugs (non-steroidal anti-inflammatory drugs (NSAIDs) or corticosteroids) are being administered to COVID-19 patients with various treatment regimens [17,18]. However, debates exist regarding their clinical use in COVID-19 patients [19,20]. For instance, ibuprofen, an over-the-counter medication used for the treatment of pain and fever in COVID-19, has been found to increase ACE2 levels [21]. In terms of corticosteroids, a recent study showed that low-dose dexamethasone, particularly in critically ill COVID-19 patients (i.e., ICU-hospitalized patients with respiratory distress), significantly improved patient survival [22]. Nevertheless, it may disrupt the immunocompetence in COVID-19 patients [23,24,25].

Histamine and its receptors play an important role in the progression of various allergic diseases [26]. Notably, the histamine H1 receptor (H1 receptor) has been reported to regulate allergic lung responses; therefore, its antagonists have been used to treat airway inflammation [27]. Beyond its role in mediating airway inflammation, our recent experimental work has identified that deptropine, a classical H1 receptor antagonist used to treat asthmatic symptoms, potently inhibits hepatitis E virus replication [28]. Along with our finding, a growing body of evidence also demonstrated that H1 receptor antagonists can inhibit various RNA virus infections [29,30]. In this review, we briefly summarize the novel use of H1 receptor antagonists in combating SARS-CoV-2 infection. The potential antiviral mechanisms of H1 receptor antagonists on SARS-CoV-2 are also discussed.

## 2. Drug Repurposing for COVID-19

Despite the development of antiviral medication and effective vaccination strategies, viral diseases remain a relevant threat to human health, especially as zoonotic reservoirs continuously give rise to novel variants of viruses [31]. The latest in a series of viral outbreaks, SARS-CoV-2 infection has now emerged as a key threat to human health. However, there is no specific approved treatment for the disease. Drug repurposing (also known as repositioning or rediscovery) has sparked extensive interest to identify medications that can be readily used for COVID-19. This strategy speeds up the conventional drug development process by testing a drug for a medical application that is different from its original indications.

To date, several nucleotide analogues have been repurposed for the treatment of COVID-19 patients (Table 1), including the nucleotide analogue remdesivir, favipiravir, ribavirin, and sofosbuvir. Treatment with intravenous remdesivir on day 7 of hospital admission markedly improved the health situation to the next day in a 35-year-old COVID-19 patient in Washington, DC, USA [32]. A Chinese open-label, controlled study indicated that favipiravir, a pyrazinecarboxamide derivative, potently inhibits SARS-CoV-2 replication [33]. Ribavirin, a broad-spectrum antiviral drug, is a guanosine analogue used for treating hepatitis C virus and hepatitis E virus, either in combination with IFNα or as a monotherapy [28,34]. Recently, ribavirin was recommended to be used with interferon α or lopinavir-ritonavir for COVID-19 treatment. However, the effects of ribavirin on COVID-19 are controversial and need to be further confirmed [35]. Recently, a molecular docking result indicated that sofosbuvir may inhibit RNA-dependent RNA polymerase (RdRP) of SARS-CoV-2 [36]. Although nucleotide analogues can efficiently inhibit viral replication by directly targeting the viral RdRP, long-term use of nucleotide analogues may exert significant off-target effects by inhibition of mitochondrial DNA polymerases [37] and may result in drug resistance [38]. Furthermore, the efficacy of direct-acting antivirals is diminishing, as evident for e.g., Hepatitis C [39] or influenza [40].

Type I interferons (IFNs) are well-characterized cytokines with potent antiviral activity which act not by directly targeting the virus but by activating the innate immunity. However, as IFNs seem to aggravate the inflammatory response in the progression to severe COVID-19 [53], the timing and period of administration for IFNs treatment needs to be considered with caution. What is more, treatment with IFNs may actually increase cellular SARS-CoV-2 entry by increasing the expression of the ACE2 receptor [54,55].

Hydroxychloroquine has been used to treat malaria and some autoimmune disorders. It was shown that hydroxychloroquine potently inhibited SARS-CoV-2 in an African green monkey kidney cell model. However, a recent study showed that hydroxychloroquine barely exerts antiviral activity in Calu-3 cells, a lung-derived epithelial cell line [46]. Furthermore, hydroxychloroquine failed to exert an anti-SARS-CoV-2 effect in a model of reconstituted human airway epithelium and in human clinical trials [56,57]. Besides the ineffectiveness, overuse of hydroxychloroquine could possibly result in multiple tissue injuries in the retina, liver, and cardiac muscle cells due to the cells’ lysosomal affinity [58,59]. Therefore, high-dose hydroxychloroquine was not recommended for COVID-19 patients, either alone or in combination with other antivirals.

Since the onset of the pandemic, lopinavir/ritonavir has been extensively used to treat COVID-19 patients [60]. However, recent studies showed that lopinavir-ritonavir had little positive effect in hospitalized COVID-19 patients [43,61]. In addition to lopinavir/ritonavir, umifenovir and oseltamivir have also been used urgently in COVID-19 patients. However, the efficacy of these drugs for COVID-19 remains debatable, particularly considering that most recent clinical trials failed to demonstrate benefits of using these drugs for COVID-19 [62,63].

The application of low molecular weight heparin (LMWH) stems from the fact that some COVID-19 patients have been reported to experience lung thromboembolism, which may progress to severe damage such as lung fibrosis in the later stages of the disease. LMWH is recommended to be used for the prevention of prophylaxis of thromboembolism in patients with severe COVID-19. Mechanistically, the cell entry step of SARS-CoV-2 is believed to be dependent on the interaction between viral spike protein and heparan sulfate. The interaction can be inhibited by heparin or synthetic heparin-like drugs, blocking viral entry. The adverse effect of using heparin in the treatment of COVID-19 patients has also been reported. For instance, the use of heparin is found to be associated with a 10–15% risk of significant bleeding [52].

Tocilizumab has been reported to improve survival and other clinical outcomes in hospitalized COVID-19 patients with hypoxia and systemic inflammation [64]. However, a recent randomized patient study showed that tocilizumab was unable to improve patient survival, although it could reduce the likelihood of developing a composite outcome of mechanical ventilation [65]. Baricitinib, an inhibitor of the Janus-associated kinases (JAKs) isoforms 1 and 2 (JAK1 and JAK2), also showed an excellent inhibitory effect on SARS-CoV-2. In a clinical trial conducted in March–April 2020 on 601 patients, 83 patients with moderate-severe SARS-CoV-2 pneumonia were treated with baricitinib. The study showed that baricitinib reduced the mortality by 71% in a large elderly cohort [50]. Most recently, a screen of ~3000 FDA-approved drugs revealed that cyclosporine showed a micromolar IC50 against SARS-CoV-2 in both Huh7.5 (IC50: 0.87 μM) and Calu-3 cells (IC50: 3.7 μM) [46].

## 3. Histamine and COVID-19

Mast cells play a crucial role in preventing parasitic, bacterial, and viral infections. Interestingly, two major host factors for SARS-CoV-2 entry, the ACE2 receptor and TMPRSS2, were also expressed on mast cells [12]. The SARS-CoV-2 infection activates mast cells, leading to the release of histamine and other inflammatory mediators [13]. Histamine is an endogenous biogenic amine that is abundant in the lungs, skin, and gastrointestinal tract, and mediates the inflammatory reaction. Histamine receptors belong to the class of G-protein coupled receptors (GPCRs), and four different histamine receptors (H1 receptor, H2 receptor, H3 receptor, and H4 receptor) have been identified [26]. H1 receptor is found to be expressed in the peripheral tissues as well as the central nervous system, mediating several allergic responses [26]. H2 receptor is highly expressed in the gastric tissues, stimulating the secretion of gastric acid and mediating gastrointestinal motility [66]. H3 receptor is mainly expressed in the central nervous system and, to a lesser extent, in the peripheral nervous system where it controls the release of a variety of neurotransmitters [67]. H4 receptor is abundantly found in bone marrow and white blood cells where it regulates mast cell chemotaxis [26].

Up to 20% of COVID-19 patients develop a severe clinical course as a result of an exaggerated inflammatory reaction in the lung. The exaggerated lung inflammation has been associated with histamine release during SARS-CoV-2 infection [13] (Figure 2A). H1 receptor is a Gαq/11 receptor containing seven membrane-spanning domains and an extracellular NH2 terminal glycosylated domain. Binding of histamine to H1 receptor has been identified to regulate lung inflammation [68]. In humans, Th2 cytokine levels are closely related to the inflammatory reaction observed in asthma. A previous study showed that allergen-challenged H1R^−/−^ mice had lower Th2 cytokine levels in the lungs compared to WT mice, suggesting a critical role of H1R-mediated Th2 cell recruitment to sites of allergic inflammation [68]. Moreover, it was found that histamine could enhance the Th1-type response through the H1 receptor; however, inhibition of Th1 and Th2 responses by histamine was observed via H2 receptors. Therefore, H1 receptor blockade could suppress the release of interferon gamma and may be effective in suppressing inflammation caused by the SARS-CoV-2 infection [69].

## 4. Inhibition of SARS-CoV-2 by H1 Receptor Antagonists

### 4.1. In Vitro and In Silico Study

Emerging evidence has reported that H1 receptor antagonists show significant promise as anti-SARS-CoV-2 medications. An in silico study has found that mizolastine exhibits excellent binding affinities when docked against the SARS-CoV-2 protease M^pro^ crystal structure [70].Doxepin, a H1 receptor antagonist, could prevent a SARS-CoV-2 spike pseudovirus from entering the ACE2-HEK293T cell with a reduced infection rate of 25.82% [71]. The antiviral effect of doxepin may rely on its inhibition of the reuptake of serotonin and norepinephrine [72], which needs to be further explored. Hydroxyzine, diphenhydramine, and azelastine were shown to exert direct antiviral activity on SARS-CoV-2 in a model in which lentivirus pseudotyped with the SARS-CoV-2 surface glycoprotein was used to infect ACE2-expressing HEK293 cells [73]. Approved drugs, such as clemastine and cloperastine, were also shown to inhibit SARS-CoV-2 in Vero cells potently [51]. Ebastine and mequitazine inhibited SARS-CoV-2 in Vero cells with IC50 values of 6.92 and 7.28 µM, respectively [74]. Of note, the antiviral activity of ebastine was cell-type-dependent, with an anti-SARS-CoV-2 effect ten-fold less in Vero cells than in Huh7.5 cells [46]. Loratadine is a H1 receptor antagonist prescribed for the treatment of hay fever and other allergic diseases. Recently, loratadine was reported to inhibit SARS-CoV-2 with an IC50 of 15.13 µM in Caco-2 cells [75]. Desloratadine could prevent SARS-CoV-2 pseudotyped virus entry in ACE2-overexpressing HEK293T cells, and the antiviral effect of desloratadine was stronger compared to loratadine [76]. Moreover, clemizole hydrochloride was found to inhibit SARS-CoV-2 (strain BavPat1) in Caco-2 cells [77]. Promethazine and terfenadine have been suggested to block SARS-CoV-2 endocytosis in respiratory epithelium [78]. A second-generation H1 receptor antagonist, rupatadine, has also been repurposed for COVID-19 [79,80]. More studies are needed to confirm the antiviral efficacy of H1 receptor antagonists on SARS-CoV-2 in lung epithelial cells (Figure 2).

### 4.2. Patient-Level Study

Recent epidemiological studies have reported that patients taking H1 receptor antagonists are more resistant to SARS-CoV-2 infection [73]. Moreover, an observational study in the Tarragona region of Spain (79,083 people over the age of 50) showed that SARS-CoV-2 infection (PCR positive rate) risk in the H1 receptor antagonists’ group is significantly lower [85]. Besides that, treatment with H1 receptor antagonists, plus azithromycin, could prevent progression to severe disease in elderly patients infected with SARS-CoV-2 [86]. Cetirizine hydrochloride is a second-generation H1 receptor antagonist. Recently, evidence showed that blocking H1/H2 receptors with cetirizine and famotidine alleviates pulmonary symptoms in COVID-19 patients. The alleviation of pulmonary symptoms is presumably attributed to their roles in minimizing the histamine-mediated cytokine storm [87]. Besides cetirizine, a case report showed that treatment with loratadine (10 mg daily) along with topical triamcinolone 0.1% cream completely resolved the rash, an initial symptom of COVID-19 infection in a 54-year-old woman [88]. More completed and ongoing clinical trials are listed in Table 2.

The above evidence together indicates that repositioning of H1 receptor antagonists represents a promising tool to provide novel therapies in the current situation. Moreover, considering the common and safe use of H1 receptor antagonists, they may be attractive prophylactic candidates for lowering the risk of SARS-CoV-2 infection in the general population.

## 5. The Role of Other Histamine Receptor Antagonists

Similar to H1 receptor antagonists, H2 receptor antagonists could also inhibit various viruses, including HCV and HIV [89]. Molecular docking results showed that H2 receptor antagonists (cimetidine, famotidine, nizatidine, and ranitidine) could bind to SARS-CoV-2 structural proteins. Moreover, the results found that famotidine and cimetidine are more prone to bind the viral proteins compared to nizatidine and ranitidine. Further, kinase enrichment analysis predicted that genes such as ERKs, SMADs, and MAPKs are involved in the antiviral activity of famotidine and cimetidine against SARS-CoV-2 [89]. In addition to the in-silico modeling, a previous retrospective cohort study has shown that treatment with famotidine was highly associated with reduced risk of mortality [90]. Another retrospective study is ongoing to evaluate the effect of these H2 receptor antagonists on the positivity rates and clinical outcomes of COVID-19 (NCT04834752). Recently, a non-randomized, un-controlled clinical trial suggested that treatment with cetirizine in combination with famotidine was effective in a favorable prognosis [87]. In fact, there are inconsistent results regarding the efficacy of famotidine against SARS-CoV-2 infection. A case study in hospitalized COVID-19 patients showed that use of famotidine relieves the clinical symptoms of COVID-19 overnight [14]. However, an in vitro study found that famotidine could not inhibit SARS-CoV-2 infection in a human intestinal organoid model derived from pluripotent stem cells [91]. Of note, improvement in COVID-19 symptoms has been associated with high-dose oral famotidine. Ten COVID-19 patients were treated with high-dose oral famotidine for 11 days (80 mg three times daily). After treatment, all patients exhibited improved clinical manifestation of COVID-19, including increased peripheral oxygen saturation [92]. However, famotidine was recommended to be given intravenously due to its poor gastrointestinal absorption and volume of distribution. In addition, although famotidine is commonly used as a safe drug in many countries, increased delirium was observed while taking this drug [93]. Taken together, the efficacy and route of administration of famotidine in COVID-19 patients needs to be further assessed. Besides H2 receptor, H4 receptor was also identified as a potential target for COVID-19 [94]. The combination of H1 and H4 receptor antagonists has been suggested to be more effective in reducing lung inflammation caused by SARS-CoV-2 [90].

## 6. Potential Anti-SARS-CoV-2 Mechanisms of H1 Receptor Antagonists

Preclinical data suggest that H1 receptor antagonists limit viral entry of Ebola/Marburg [95] or Hepatitis C virus (HCV) [96] even if the molecular pathways explaining these antiviral effects remain largely obscure. For example, diphenyl-piperazines (e.g., chlorcyclizine, cyclizine, and hydroxyzine), cycloheptene-piperidines (e.g., cyproheptadine, ketotifen, loratadine, and desloratadine), and phenothiazines (e.g., mequitazine and trimeprazine) have been found to inhibit HCV cell entry. Interestingly, the potential anti-HCV mechanisms of these drugs are likely independent of H1 receptor [97].

### 6.1. Intervention of Early Step of SARS-CoV-2 Infection

Consistent with HCV and Ebola, the H1 receptor antagonists may also inhibit SARS-CoV-2 infection via intervention in the early step of virus replication. It may prevent SARS-CoV-2 from entering the cell host through binding ACE2 [73]. Moreover, evidence showed that H1 receptor antagonists could inhibit SARS-CoV-2 infection through binding sigma receptor-1 (for hydroxyzine and diphenhydramine), an endoplasmic reticulum resident chaperone protein participating in early steps of SARS-CoV-2 replication [51,73]. Heparin belongs to glycosaminoglycan molecules which include heparan sulphate. During virus entry, SARS-CoV-2 spike protein was demonstrated to interact with both cellular heparan sulfate and ACE2 [16]. Therefore, disrupting the interaction between heparan sulphate and spike protein by exogenous and competitive heparin mimetics could limit the SARS-CoV-2 virus entering the cell, resulting in suppression of the inflammatory responses [98]. Histamine has been shown capable of binding to heparan sulfate [99,100]. It is interesting to investigate the involvement of heparan sulfate in the H1 receptor antagonist-mediated anti-SARS-CoV-2 activity.

Unlike H1 receptor, the proposed anti-SARS-CoV-2 mechanisms for H2 receptor antagonists are inhibition of Type 2 transmembrane serine protease (TMPRSS2) and 3-chymotrypsin-like protease (3CLpro) by famotidine [101]. However, a recent study revealed that the principal mechanism of action of famotidine for relieving COVID-19 symptoms is attributed to its on-target effect on H2 receptor activity, and is likely independent of other histamine receptors determined by competition assay [14]. Most recently, molecular docking results suggest that famotidine and cimetidine inhibit SARS-CoV-2 replication through the inhibition of non-structural proteins, including NSP3, NSP7/8 complex, and NSP9 [89]. The modes of action of anti-SARS-CoV-2 activity of famotidine needs to be further clarified.

### 6.2. NF-κB-Mediated Antiviral Activity by H1 Receptor Antagonist

Activation of cellular inflammation was observed following histamine binding to H1 receptor. This activation was mediated by phospholipase C (PLC) [82], protein kinase C (PKC), and NF-κB signaling pathways (Figure 2B) [102]. Our recent experimental work demonstrated that deptropine could potently inhibit HEV replication. The anti-HEV effect of deptropine requires the inhibition of NF-κB activity and is likely dispensable of H1 receptor [28]. Interestingly, activation of NF-κB in lung epithelial cells was found to facilitate SARS-CoV-2 propagation, and H1 receptor antagonists were able to suppress the activation of NF-κB in lung epithelial cells [81,84]. The above evidence together prompts us to hypothesize that the potential mechanism by which H1 receptor antagonists inhibit SARS-CoV-2 may include their inhibitory effects on NF-κB [28]. Of note, H1 receptor antagonists can inhibit NF-κB through H1 receptor-dependent and -independent mechanisms [84]. The implication of H1 receptor and the NF-κB signaling pathway in a H1 receptor antagonists-mediated anti-SARS-CoV-2 effect is intriguing and needs to be further explored.

### 6.3. Other Potential Antiviral Mechanisms

The observation that H1 receptor antagonists exert antiviral effects, however, also fits well with recent observations that the action of H1 receptor in physiology is much broader than the conventional view that such action is mainly restricted to allergic inflammation. The endogenous histamine stimulates oleoylethanolamide biosynthesis in the liver via H1 receptor activation, regulating liver ketogenesis [103]. Epidemiological evidence as well as preclinical studies indicate that H1 receptor antagonists are associated with increased food intake, body weight, and obesity [104]. The H1 receptor was also shown to play a vital role in the control of cellular metabolism. Genetic deletion of H1 receptor in mice leads to increased abdominal adiposity and glucose intolerance [105]. Metabolism and immune responses are two fundamental biological processes that protect the cell host from viral infection. Interestingly, it has been proposed that H1 receptor can modulate innate immunity beyond its role in regulating allergic reactions [106]. Thus, the blockage of H1 receptor by its antagonists may affect the SARS-CoV-2 replication via mediation of the crosstalk between metabolism and immune responses.

## 7. Opportunities and Challenges

There are various advantages in the application of H1 receptor antagonists for the treatment of COVID-19. First, these drugs are relatively inexpensive and readily to be used. Second, although the chronic inflammation may not be the primary driver of pathology, using H1 receptor antagonists can improve patient outcomes due to its natural role in reducing inflammation. However, it also should be noted that the first-generation H1 receptor antagonists are not strictly selective for the H1 receptor. They contribute to a spectrum of dopaminergic and cholinergic reactions, resulting in significant adverse effects in the central nervous system due to their capability of penetrating the blood-brain barrier [107]. Moreover, they are able to regulate cardiac channels, leading to wide range of side effects, including sedation, insomnia, and hyperactivity [108]. Besides that, H1 receptor antagonists have been associated with an increase in seizure susceptibility [109]; therefore, H1 receptor antagonists in most cases are not recommended to be prescribed to COVID-19 patients with epilepsy or febrile seizures. Another factor that may hinder the clinical use of H1 receptor antagonists in COVID-19 is the impairment of innate immune responses by these drugs [110].

As mentioned above, both loratadine and desloratadine could inhibit SARS-CoV-2 infection. They are new-generation H1 receptor antagonists which are highly selective to H1 receptor with less central nervous system side effects. Besides their role in reducing lung inflammation induced by histamine, they can suppress a number of other inflammatory activities [76], which make them ideal medications for relieving the excess inflammatory response of COVID-19. Overall, with studies describing H1 receptor antagonists to be capable of treating viral diseases, recent findings take us one step closer to understanding the underlying antiviral mechanism and the advantages of the H1 receptor antagonists in the treatment of COVID-19. In the future, the use of H1 receptor antagonists as antiviral agents will further advance our understanding of how these agents work beyond their intrinsic effects on the histamine receptor, and bear potential implications for the real-world treatment of COVID-19 patients.

## Figures and Tables

**Figure 1 ijms-22-05672-f001:**
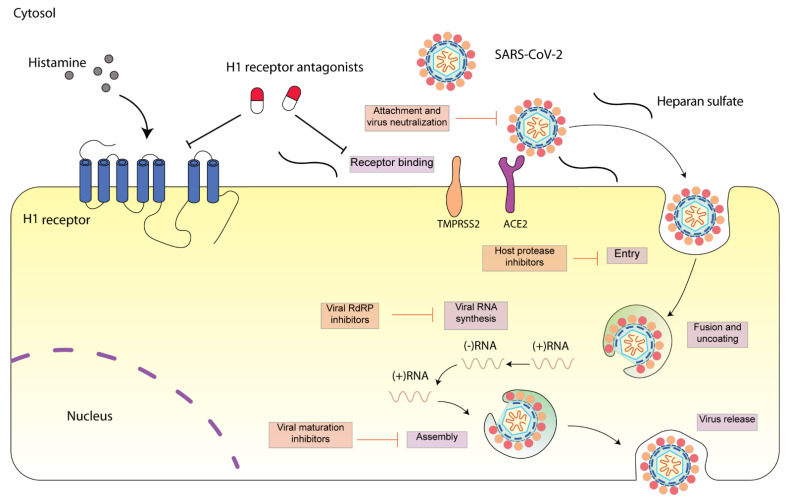
Schematic diagram presenting life cycle of SARS-CoV-2 and relevant inhibitors. SARS-CoV-2 cell entry begins with binding of the spike S protein to ACE2, a process that is facilitated by TMPRSS2. SARS-CoV-2 enters the cell through endocytosis, and then the virus is uncoated in the acidic environment of lysosomes. After that, SARS-CoV-2 RNA is released, followed by the reproduction of virus genome and viral proteins. Then, the viral components are assembled and released via exocytosis [15]. Each step can be targeted by relevant inhibitors. H1 receptor antagonists may inhibit SARS-CoV-2 either via H1 receptor or via ACE2 receptor. SARS-CoV-2 spike protein interacts with both cellular heparan sulfate and ACE2 through its receptor-binding domain (RBD) [16]. H1 receptor antagonists may disrupt the interaction between heparan sulfate and spike protein, inhibiting SARS-CoV-2 entry.

**Figure 2 ijms-22-05672-f002:**
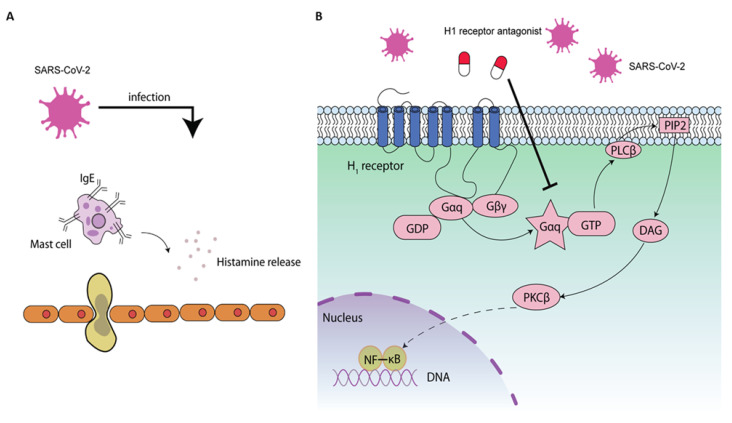
The potential antiviral mechanisms of H1 receptor antagonists on SARS-CoV-2. SARS-CoV-2 infection leads to histamine release (**A**) and activates the NF-κB signaling (**B**), leading to enhancement of inflammatory response to facilitate its replication [81]. Histamine-mediated NF-κB signaling has been associated with the upstream phospholipase C (PLC) and protein kinase C (PKC) activation [82]. The activation of NF-κB by histamine can be blocked by H1 receptor antagonists [83]. H1 receptor antagonists may inhibit the NF-κB signaling through H1 receptor-dependent or -independent mechanisms [84].

**Table 1 ijms-22-05672-t001:** Mode of action of several potential antiviral drugs repurposed for COVID-19.

Drug	Mode of Action	References
Remdesivir	Binds to the viral-RNA dependent RNA polymerase (RdRp), terminating transcription of viral RNA	[41]
Hydroxychloroquine	Increases the endosomal pH, suppressing the fusion of SARS-CoV-2 with the host cell membrane	[42]
Lopinavir/Ritonavir	Inhibits the protein 3CLpro, required for cleaving poly protein into RNA dependent RNA polymerase and helicase	[43]
Umifenovir	Blocks the fusion of virus to the cell/endosome by interfering with the hydrogen bond network in the phospholipid	[44]
Favipiravir	Destroys the conservative catalytic domain of RdRp, interrupting the nucleotide incorporation	[45]
Cyclosporine	Targets cyclophilin rather than calcineurin	[46]
Recombinant human ACE2	Blocks virus cell entry	[47]
Oseltamivir	Interferes with viral exocytosis	[48]
Sofosbuvir	Binds to the viral-RNA dependent RNA polymerase (RdRp)	[49]
Valinomycin	Inhibits S-phase kinase-associated protein	[47]
Baricitinib	Targets both viral entry and the cytokine storm	[50]
Zotatifin	Inhibits cap-dependent mRNA translation through the host translationmachinery	[51]
Heparin	Inhibits viral spike protein-cell receptor interaction and block viral entry	[16,52]

**Table 2 ijms-22-05672-t002:** Examples of histamine (H1/H2) receptor antagonists repurposed for COVID-19.

Drug	Doses	Clinical Studies
Cyproheptadine	4 mg three times a day for 10 days	NCT04876573
Famotidine & N-Acetyl Cysteine	N-Acetyl Cysteine 600~1800 mg three times daily; Famotidine 20~80 mg three times daily	NCT04545008
Celecoxib & Famotidine plus Remdesivir	80 mg four times daily for 7 days and then 40 mg twice daily for a course of 14 days	NCT04488081
Cetirizine & Famotidine	10 mg of cetirizine once a day and 20 mg of famotidine twice a day for 10 days	NCT04836806
Famotidine	80 mg three times a day for a maximum of 14 days	NCT04724720

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
