# Peer review of "Could Histamine H1 Receptor Antagonists Be Used for Treating COVID-19?"

_ijms, 2021, doi:10.3390/ijms22115672_

Round 1
Reviewer 1 Report
This paper addresses a very interesting question, namely could H1 receptor antagonists be used to help treat COVD-19. Indeed in April 2020, Michael Johnson described this in a research proposal ( https://www.researchgate.net/publication/340575298_Histamine_as_a_Potential_Therapeutic_Target_for_Preventing_COVID-19_Progression_to_ARDS). I believe that the paper would benefit from sticking to the authors’ main question and removing some of the text.
Line 79: I do not believe that histamine antagonists are widely used to treat asthma. I refer the authors to the guidelines from the British Thoracic Society which state that “Antihistamines and ketotifen are ineffective”. Please see https://www.brit-thoracic.org.uk/quality-improvement/guidelines/asthma/ BTS/SIGN Guideline for the management of asthma 2019 and the cited reference: Van Ganse E, Kaufman L, Derde MP, Yernault JC, Delaunois L, Vincken W. Effects of antihistamines in adult asthma: a meta-analysis of clinical trials. Eur Respir J 1997;10(10):2216-24.
Line 48: Reference 8 does not have any information about mast cells in the lung. Reference 9 is a review article, and the authors should cite the original work (Motta Junior JDS, Miggiolaro AFRDS, Nagashima S, de Paula CBV, Baena CP, Scharfstein J, de Noronha L. Mast Cells in Alveolar Septa of COVID-19 Patients: A Pathogenic Pathway That May Link Interstitial Edema to Immunothrombosis. Front Immunol. 2020 Sep 18;11:574862. doi: 10.3389/fimmu.2020.574862. PMID: 33042157; PMCID: PMC7530169). The original work states: “Here we report histopathology data obtained in post-mortem lung biopsies of COVID-19, showing the increased density of perivascular and septal mast cells (MCs) and IL-4-expressing cells (n = 6), in contrast to the numbers found in pandemic H1N1-induced pneumonia (n = 10) or Control specimens (n = 10).”
Figures 1 and 2 lack any references in the legend
The section lines 97-156 and Table 1 be removed, as they do not add to the substance of this review. They could then discuss drug repurposing studies about H1 receptor antagonists e.g. Glebov OO. Understanding SARS-CoV-2 endocytosis for COVID-19 drug repurposing. FEBS J. 2020 Sep;287(17):3664-3671. doi: 10.1111/febs.15369. Epub 2020 Jun 2. PMID: 32428379; PMCID: PMC7276759; Tachoua W, Kabrine M, Mushtaq M, Ul-Haq Z. An in-silico evaluation of COVID-19 main protease with clinically approved drugs. J Mol Graph Model. 2020 Dec;101:107758. doi: 10.1016/j.jmgm.2020.107758. Epub 2020 Sep 21. PMID: 33007575; PMCID: PMC7503128; Touret F, Gilles M, Barral K, Nougairède A, van Helden J, Decroly E, de Lamballerie X, Coutard B. In vitro screening of a FDA approved chemical library reveals potential inhibitors of SARS-CoV-2 replication. Sci Rep. 2020 Aug 4;10(1):13093. doi: 10.1038/s41598-020-70143-6. PMID: 32753646; PMCID: PMC7403393, as well as including other papers that they cite.
Lines 221-222. Reference 72 says nothing about cetirizine and patients indeed it stated “Currently, we could not find studies evaluating the efficacy of H1R blockers in COVID-19.” Furthermore reference 73 is about the measurement of “cetirizine hydrochloride in the presence of two antimalarials” and not about a clinical trial. Although it does contain the sentence “cetirizine hydrochloride and fexofenadine combination are currently being tested in clinical trials for COVID-19” (almost identical to the sentence in this manuscript. This refers to their reference 27 “Courtney A. J. New COVID-19 clinical trial will utilize combination of two historically safe drugs. https://www.wlox.com/, June 9, 2020 at 7:15 PM CDT - Updated June 10, (accessed June 18, 2020).”.
The authors could check clinicaltrials.gov for details of trials involving histamine antagonists.
Lines 258-262: the authors seem to contradict themselves first saying: that the drugs inhibit HCV cell entry by acting as H1 receptor antagonists and then saying the potential anti-HCV mechanisms are likely independent of the H1 receptor.
References need to be thoroughly checked as they are presented very inconsistently. I have only looked at this section very quickly, so this list of corrections is not exhaustive!! In some references the journal titles are given in full, whilst others use the recognized abbreviations. The use of capitalization also needs to be checked carefully both in journal titles and the title of the article. Authors names are occasionally also given in capitals e.g. reference 11. References are not always complete e.g. references 12, 60, 66, 72, 80, 83. Reference 41 is not correctly cited: Dittmar M, Lee JS, Whig K, Segrist E, Li M, Kamalia B, Castellana L, Ayyanathan K, Cardenas-Diaz FL, Morrisey EE, Truitt R, Yang W, Jurado K, Samby K, Ramage H, Schultz DC, Cherry S. Drug repurposing screens reveal cell-type-specific entry pathways and FDA-approved drugs active against SARS-Cov-2. Cell Rep. 2021 Apr 6;35(1):108959. It is also the same as reference 55! Reference 48 has no authors listed; it should read: RECOVERY Collaborative Group. References 12 and 56 are identical. Where were references 59 and 67 published?
Minor changes
Line 48 remove reference 8 and substitute reference 9 with the original reference. Also alveolar is an adjective and the original work refers to: density of perivascular and septal mast cells
Lines 49-50, I would suggest that the reference by Malone et al., be add in here (reference 90)
Line 50 severe not sever
Line 65 consequence not consequent
Lines 97-110: the font size has changed
Lines 161-164: Please provide references for the statement that SARS-CoV-2 infection activates mast cells. Also when correcting it should be activates not activate.
Lines 176-177: please provide references for the statement that the exaggerated lung inflammation is associated with the release of histamine during SARS-CoV-2 infection.
Line 192: evidence not evidences
Lines 193-194 directly copied from the article “doxepin could inhibit SARS-CoV-2 spike pseudovirus from entering the ACE2-expressing cell, reducing the infection rate to 25.82%”. In your paper you will need to say what sort of cell is under investigation i.e. ACE2-HEK293T cells.
Lines 195-196 directly copied from the article: “ Hydroxyzine, diphenhydramine and azelastine exhibited direct antiviral effects against SARS-CoV-2”. Again in your paper you need to state what they do.
Line 225: refence 73 does not seem to relate to what has been written.
Line 229 evidence not evidences
Lines 292-293 – please replace reference 94 with the original paper
Line 195: evidence not evidences
Reviewer 2 Report
The manuscript ijms-1210818: “Could histamine H1 receptor antagonists be used for treating COVID-19?“ presents an important literature review focused on finding therapeutic solution for fighting SARS-CoV-2 infections.
I analyzed the manuscript and added here a few suggestions in order to help the authors improve their paper.
I think the authors should present relevant information from the following articles:
Histamine receptors and COVID-19, Inflamm Res. 2020 Nov 18 : 1–9.
COVID-19: Famotidine, Histamine, Mast Cells, and Mechanisms, Front. Pharmacol., 23 March 2021 | https://doi.org/10.3389/fphar.2021.633680
The section 5 should be extended considering the volume of data on antiH2 drugs and their potential role in COVID-19 treatment.
The section 97-110 needs to be edited as the journal requirements. It should be expended to present more examples of repurposed drugs analyzed as COVID-19 treatment. See the articles:
Comprehensive analysis of drugs to treat SARS‑CoV‑2 infection: Mechanistic insights into current COVID‑19 therapies (Review), Int J Mol Med. 2020 Aug;46(2):467-488
A SARS-CoV-2 protein interaction map reveals targets for drug repurposing, Nature, 583, pages459–468(2020).
Row 193, the section “Doxepin, a H1 receptor antagonist” needs correction. The authors should add that doxepin inhibits the reuptake of serotonin and norepinephrine and has antiadrenergic and anticholinergic activities. The reader should know that these effects may play also a role in its antiviral effect.
A new section should be added to detail the clinical studies performed or currently underway using antiH1 drugs in COVID-19 patients. A table should be added presenting the drug used, the clinical assay code, doses and administration, and if available, clinical results.
The section 254-262 describes other viruses, and not SARS-CoV-2. Please check and correct.
Round 2
Reviewer 1 Report
The authors have addressed most of the comments raised in the original review.
There are still some issues with the language, which need editorial correction
In the abstract (line 17) the authors say that H1 antagonists are widely used to treat lung inflammation, this is not correct.
The reference section still needs further work. Many things that should have a capital letter are in lower case e.g. reference 1 wuhan, china should be Wuhan, China. There are too many mistakes to list here.
Lines 70-71 the authors use heparan on line 70 but heparin on line 71. Which is correct?
Lines 92-92 need the reference to the author's work that they mention
Reviewer 2 Report
The authors performed important modifications to their paper and largely improved the quality of the paper. I consider that it can be published after small editorial corrections.
